# Loss of Doc2b does not influence transmission at Purkinje cell to deep nuclei synapses under physiological conditions

Mehak M Khan, Wade G Regehr*

Department of Neurobiology, Harvard Medical School, Boston, United States

**Abstract** Doc2a and Doc2b are high-affinity calcium-binding proteins that interact with SNARE proteins and phospholipids. Experiments performed on cultured cells indicated that Doc2 proteins promote spontaneous vesicle fusion and asynchronous neurotransmitter release, regulate vesicle priming, mediate augmentation, and regulate transmission during sustained activity. Here, we assess the role of Doc2 proteins in synaptic transmission under physiological conditions at mature synapses made by Purkinje cells onto neurons in the deep cerebellar nuclei (PC to DCN synapses). PCs express Doc2b but not Doc2a. Surprisingly, spontaneous neurotransmitter release, synaptic strength, the time course of evoked release, responses evoked by sustained high-frequency stimulation, and short-term plasticity were normal in Doc2b KO mice. Thus, in stark contrast to numerous functions previously proposed for Doc2, here we find that Doc2b removal does not influence transmission at PC-to-DCN synapses, indicating that conclusions based on studies of Doc2b in cultured cells do not necessarily generalize to mature synapses under physiological conditions.

*For correspondence:
wade_regehr@hms.harvard.edu

**Competing interests:** The authors declare that no competing interests exist.

## Introduction

Presynaptic calcium signaling plays a critical role in neurotransmitter release and synaptic plasticity. Numerous types of Ca-binding proteins are present in presynaptic terminals and have the potential to contribute to synaptic transmission. Of particular interest are proteins that contain tandem C2 domains that bind Ca and interact with phospholipids and SNARE proteins, such as several Ca-sensitive Synaptotagmin isoforms (Syts) and two Doc2 isoforms, Doc2a and Doc2b (*Groffen et al., 2010*; *Pang et al., 2011*; *Yao et al., 2011*). Some of these proteins (Syt1 and Syt2) bind calcium with low affinity and fast kinetics to mediate fast synaptic transmission (*Fernández-Chacón et al., 2001*). Others, such as Syt7, bind Ca with high affinity and slow kinetics and mediate facilitation and asynchronous neurotransmitter release (*Wen et al., 2010*; *Jackman et al., 2016*). Doc2b and Doc2a also bind calcium with high affinity, but their contribution to synaptic transmission is unclear.

Multiple physiological functions have been ascribed to Doc2a and Doc2b based primarily on cell culture studies. These proteins are understood to function in a similar manner but can be differentially expressed (*Courtney et al., 2018*; *Verhage et al., 1997*). In hippocampal cultures, Doc2b knockout (KO) reduced the frequency of miniature postsynaptic currents ('minis') compared to control (*Groffen et al., 2010*; *Courtney et al., 2018*). This reduction in mini frequency has also been observed at Purkinje cell (PC) to PC collateral synapses in acute slices from P7-8 Doc2b KO mice (*Groffen et al., 2010*). These results led to the proposal that Doc2b is a Ca sensor for spontaneous release. Doc2b and Doc2a have also been proposed to be Ca sensors for asynchronous release in hippocampal cultures (*Yao et al., 2011*). Another study from hippocampal cultures reported that Doc2 proteins mediate synaptic augmentation (*Xue et al., 2018*). Furthermore, in cultured

chromaffin cells, Doc2b is thought to act as a Ca sensor for vesicle priming (*Houy et al., 2017*). Despite the intriguing results of these studies, the role of Doc2b at mature synapses under physiological conditions remains unaddressed.

Here, we investigate how Doc2b contributes to transmission at the PC to Deep Cerebellar Nuclei (DCN) synapse in mature animals. PCs form powerful synapses onto neurons in the DCN which relay cerebellar output to numerous brain regions. This synapse was chosen because *Doc2b* is expressed in PCs but *Doc2a* is not (*Verhage et al., 1997*; *Groffen et al., 2010*), and because Doc2b might contribute to several distinguishing features of this synapse. The PC to DCN synapse remains effective even for high-frequency sustained activation, as is typical in vivo (*Turecek et al., 2016*; *Zhou et al., 2014*). Furthermore, a specialized vesicle pool with a very low initial probability of release is thought to mediate transmission during high-frequency activation, but this pool is poorly understood. We hypothesized that Doc2b regulates this pool because Doc2b regulates release during prolonged stimulation of chromaffin cells (*Pinheiro et al., 2013*). Second, facilitation at the PC to DCN synapse helps these synapses maintain frequency-invariance (*Turecek et al., 2017*). Although Syt7 has been proposed to mediate facilitation at the PC to DCN synapse, we tested the possibility that Doc2b also contributes to short-term synaptic plasticity and frequency-invariance. Unexpectedly, no aspect of PC to DCN transmission was altered in Doc2b KO mice. These results present a striking dichotomy between the previously described contributions of Doc2b to the synaptic physiology of cultured neurons and the lack of an apparent role in transmission at an intact synapse under physiological conditions.

## Results

### Spontaneous release

In order to assess the suitability of the PC to DCN synapse for our studies, we used fluorescence in situ hybridization (FISH) to evaluate *Doc2b* and *Doc2a* gene expression in adult (P60-P70) wildtype and Doc2b KO mice (*Groffen et al., 2010*). PCs strongly expressed *Doc2b* but not *Doc2a* in wildtype animals (*Figure 1A*). In PCs of Doc2b KO mice, *Doc2b* expression was eliminated and *Doc2a* expression remained absent (*Figure 1B*). In wildtype animals, DCN neurons did not express either *Doc2b* or *Doc2a* (*Figure 1A*). *Doc2b* was apparent in the dentate gyrus of the hippocampus (*Verhage et al., 1997*; *Figure 1—figure supplement 1A*), and the striatum (data not shown) of wildtype animals but was absent in the KO (*Figure 1—figure supplement 1B*). *Doc2a* expression was apparent in the CA3 region of the hippocampus in both WT and Doc2b KO (*Figure 1—figure supplement 1*). Furthermore, we used immunostaining to demonstrate that Doc2b protein colocalizes with parvalbumin, a marker for PC boutons, in the DCN of WT animals (*Figure 1C*). Doc2b immunoreactivity was not detected in Doc2b KO animals (*Figure 1D*). Doc2b immunoreactivity was also prominent in PC cell bodies and dendrites and in other brain regions including the hippocampus (*Figure 1—figure supplement 2*). These studies indicate that Doc2b is the only calcium-dependent Doc2 present at PC synapses of adult wildtype mice, that it is eliminated in Doc2b KO mice, and that there is no compensatory expression of *Doc2a* in PCs of Doc2b KO mice.

Based on multiple studies that show that Doc2b positively regulates mini frequency (*Courtney et al., 2018*; *Groffen et al., 2010*; *Pang et al., 2011*; *Ramirez et al., 2017*), we expected the loss of Doc2b to decrease mIPSC frequency at the PC to DCN synapse. Surprisingly, however, there was no difference in mIPSC frequency for WT and Doc2b KO mice at the mature PC to DCN synapse under physiological conditions (34-35° C, 1.5 mM external calcium, $Ca_e$) (p=0.93, two-tailed unpaired Student's *t*-test) (*Figure 1E–F*). This contrasts with the strong reduction in mIPSC frequency that Doc2b KO causes in hippocampal autapse cultures (*Groffen et al., 2010*), hippocampal neuronal cultures (*Courtney et al., 2018*; *Groffen et al., 2010*; *Pang et al., 2011*; *Ramirez et al., 2017*), and PC to PC synapses (*Groffen et al., 2010*). In the latter case, the observed 75% reduction in mIPSC frequency at PC to PC synapses is paricularly interesting because it also involves synapses made by PCs. In contrast to our PC to DCN studies which were perfomed in near physiological conditions (34-35° C, 1.5 $Ca_e$), experiments at PC to PC synapses were performed in brain slices of P7-P8 mice at room temperature in elevated $Ca_e$ (2.0 mM). This raises the possibility that the age of the animal, the temperature of the experiments, or the external calcium levels could all contribute to differences in the Doc2b dependence of mIPSC frequency. We found that mIPSC frequency was

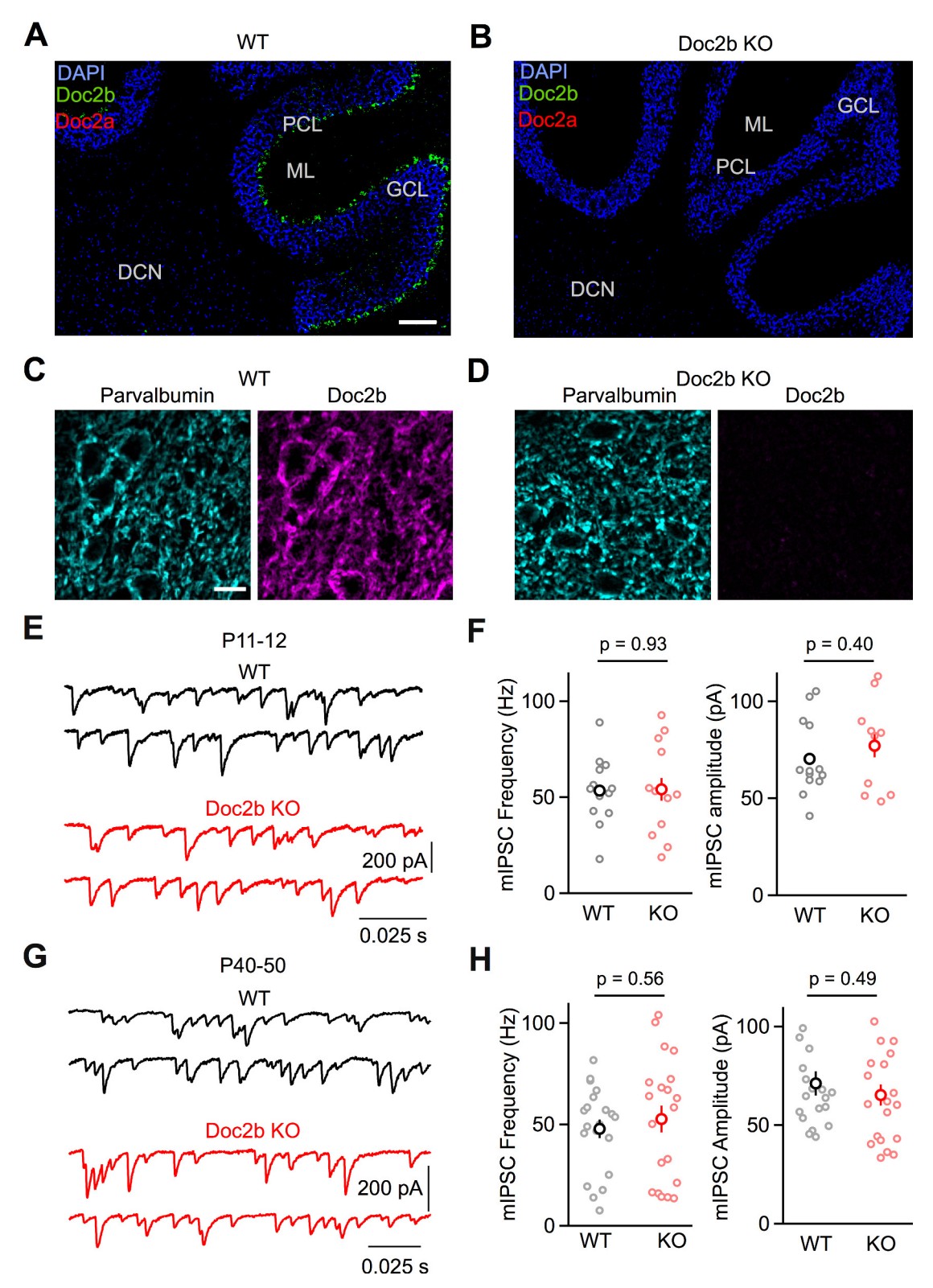

**Figure 1.** Loss of Doc2b does not affect mIPSC frequency at the PC to DCN synapse. (**A**) Sagittal cerebellum slice from a WT mouse at postnatal day 60 (P60) labeled using FISH for *DAPI* (blue), *Doc2b* (green), and *Doc2a* (red). Abbreviations, Purkinje Cell Layer (PCL), Molecular Layer (ML), Granule Cell Layer (GCL), and Deep Cerebellar Nuclei (DCN). Scale bar, 200 µm. (**B**) Same as in (**A**) but for Doc2b KO. Images are on the same scale as in (**A**). (**C**) Sagittal DCN section from a WT mouse at postnatal day 60 (P60) immunostained for parvalbumin (left, cyan) and Doc2b (right, magenta). Scale bar, 25
*Figure 1 continued on next page*

*Figure 1 continued*

μm. (D) Same as in (C) but for Doc2b KO. Images are on the same scale as in (C). (E) mIPSCs were recorded at the PC to DCN synapse in P40-50 mice in the presence of TTX, NBQX, and CPP. Representative traces from individual cells for WT and Doc2b KO littermates. (F) Summary data for mIPSC frequency and amplitude at the PC to DCN synapse in P40-50 mice. n = 14 cells (3 animals) for WT, n = 12 cells (3 animals) for Doc2b KO. Individual cells are shown as small circles and genotype average is shown as larger bold circle. Statistical significance was assessed using two-tailed Student's *t*-tests after data were found to be normally distributed (Shapiro-Wilk test) (see Materials and Methods for details). (G) Same as in (C) but for P11-12 mice. (H) Same as in (D) but for P11-12 mice. n = 20 cells (3 animals) for WT, n = 20 cells (2 animals) for Doc2b KO.

The online version of this article includes the following source data and figure supplement(s) for figure 1:

**Source data 1.** PC to DCN mini frequencies and amplitudes.
**Figure supplement 1.** *Doc2a* expression is normal in Doc2b KO.
**Figure supplement 2.** Doc2b immunohistochemistry in wildtype and Doc2b KO animals.
**Figure supplement 3.** Doc2b KO reduces mIPSC frequency at PC to PC collateral synapses.
**Figure supplement 4.** Postsynaptic loss of Doc2b does not alter mEPSC frequency onto PCs.

reduced by 65% in P7-P8 in Doc2b KO mice for experiments that were performed at 34-35° C and in 1.5 $Ca_e$ (p<0.001, two-tailed Wilcoxon signed-rank test) (***Figure 1—figure supplement 3***). This result suggests that the age of the animal could account for the Doc2b dependence of mIPSC frequency. It is impractical to extend studies of the PC to PC synapse to older animals because PCs are the dominant source of inhibition for other PCs for only a brief developmental window, and in older animals, molecular layer interneurons become the dominant source of inhibition (***Altman, 1972***; ***Bernard and Axelrad, 1993***). These observations suggest that Doc2b contributes to mini release in developing animals but not in adults. To address this possibility, we repeated experiments at the PC to DCN synapse in P11-12 animals, but again observed no difference in mini release in Doc2b KO mice (p=0.57, two-tailed unpaired Student's *t*-test) (***Figure 1G–H***).

Another potential issue in interpreting previous studies is that Doc2b is present both presynaptically and postsynaptically at PC to PC synapses and at many cultured synapses that have been studied. Although Doc2b is thought to function presynaptically, it remains possible that Doc2b may act postsynaptically to influence mini frequency. We therefore tested whether postsynaptic loss of Doc2b affects mini frequency by recording miniature excitatory postsynaptic currents (mEPSCs) at the parallel fiber to PC synapse, in which *Doc2b* is normally expressed in the postsynaptic PC, but not in the presynaptic granule cells (***Figure 1A***). At the parallel fiber to PC synapse, we saw no effect on mEPSC frequency or amplitudes in Doc2b KO mice (p=0.83), arguing against a postsynaptic contribution of Doc2b to minis (***Figure 1—figure supplement 4***).

## Synaptic strength and release kinetics

We assessed the role of Doc2b in determining the strength and kinetics of evoked release by studying individual PC inputs. To isolate synaptic responses from single PCs, we stimulated PC axons at the stimulus threshold and evoked failures and synaptic currents with a similar probability (***Figure 2A–B***). The peak single fiber conductances were similar for WT and Doc2b KO mice (p=0.22, two-tailed Wilcoxon signed-rank test) (WT: 30.9 ± 3.5 nS, n = 30 inputs, Doc2b KO: 26.9 ± 2.7, n = 29 inputs), indicating that the loss of Doc2b does not alter evoked release at the PC to DCN synapse.

The high affinity and slow kinetics of calcium-binding made Doc2 a strong candidate sensor for asynchronous release, and a study concluded that Doc2 mediates slow asynchronous release in hippocampal cultures (***Yao et al., 2011***). However, we found that the decays of the synaptic currents were unaltered in Doc2b KO mice (***Figure 2B–C***). The IPSC decay time constants were similar for WT and Doc2b KO mice (p=0.45, two-tailed unpaired Student's *t*-test) (WT: 3.43 ± 0.18 ms, Doc2b KO: 3.63 ± 0.17 ms). We also quantified asynchronous release by detecting individual inhibitory synaptic events. There was a high frequency (50–100 Hz) of spontaneous IPSCs onto DCN cells. As expected, during failure trials, there was no change in spontaneous event frequency (***Figure 2—figure supplement 1***). If asynchronous release is present at the PC to DCN synapse, the number of events after a success trial would be expected to transiently increase, but this was not the case (***Figure 2—figure supplement 1***). Instead there was a transient decrease in sIPSC frequency. This reduction was present in both WT and Doc2b KO mice and could be lengthened by stimulating PCs 4 times at 100 Hz (***Figure 2—figure supplement 2***). Several possible mechanisms could account for

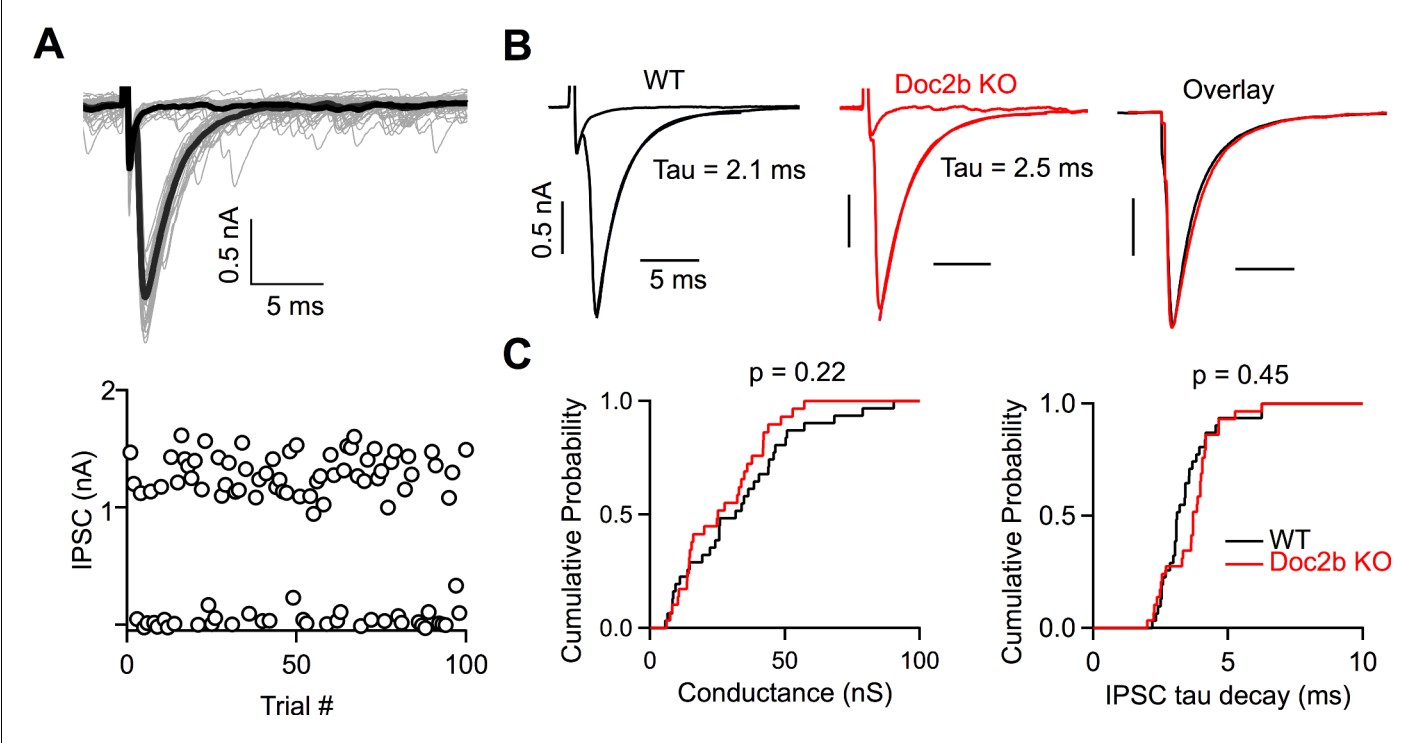

**Figure 2.** Doc2b does not contribute to the strength or kinetics of evoked release. (**A**) Top: Example minimal stimulation of a PC input to a DCN neuron, where the stimulus intensity was lowered until failures or single inputs were evoked with similar probability. Bold traces show average traces for failures and single inputs. Bottom: IPSC amplitudes at a fixed stimulus intensity. (**B**) Representative examples of single inputs from WT and Doc2b KO mice. (**C**) Summary data for minimal conductances and single IPSC decay time constant (tau) for WT and Doc2b KO. n = 30 inputs for WT, n = 29 inputs for Doc2b KO. Average conductance was 30.9 ± 4 nS for WT and 26.9 ± 3 nS for Doc2b KO. Average IPSC tau decay was 3.43 ± 0.2 ms for WT and 3.63 ± 0.2 for Doc2b KO. Left: Statistical significance was assessed using two-tailed Wilcoxon Signed-Rank test after the data were found to be asymmetrically distributed (Shapiro-Wilk test). Right: Statistical significance was assessed using two-tailed Student's t-tests after data were found to be normally distributed (Shapiro-Wilk test).

The online version of this article includes the following source data and figure supplement(s) for figure 2:

**Source data 1.** PC to DCN single fibers and spontaneous events.

**Figure supplement 1.** Asynchronous release after single stimuli does not occur at the PC to DCN synapse.

**Figure supplement 2.** Asynchronous release after stimulus bursts does not occur at the PC to DCN synapse.

the transient decrease of sIPSC frequency, including depletion of the readily releasable pool, or shunting of more distal inputs as somatic PC conductances are activated. However, with regard to the issue of asynchronous release, these experiments demonstrate that asynchronous release is not detected at the PC to DCN synapse in WT or Doc2b KO animals.

## Sustained responses and synaptic plasticity

PCs usually fire at high frequencies for sustained periods, and it is therefore important to use appropriate stimulus patterns to assess the contribution of Doc2b under physiological conditions. The PC to DCN synapse changes considerably during development. We therefore examined the role of Doc2b at PC to DCN synapses in both juveniles (P13-15) and adults (P60-80). We tested the hypothesis that Doc2b helps PC to DCN synapses maintain their efficacy during sustained periods of high-frequency activation.

We began by studying the PC to DCN synapse in juveniles, where synaptic depression is prominent and becomes more pronounced at higher PC stimulation frequencies (*Figure 3*; *Turecek et al., 2017*; *Turecek et al., 2016*). We stimulated PC axons with 50 stimuli delivered at frequencies between 5 Hz and 100 Hz and recorded synaptic responses in DCN cells. In both WT and Doc2b KO mice, frequency-dependent synaptic depression was readily observable, and responses from individual cells were comparable (*Figure 3A–B*). Paired pulse plasticity was unaffected by the loss of Doc2b

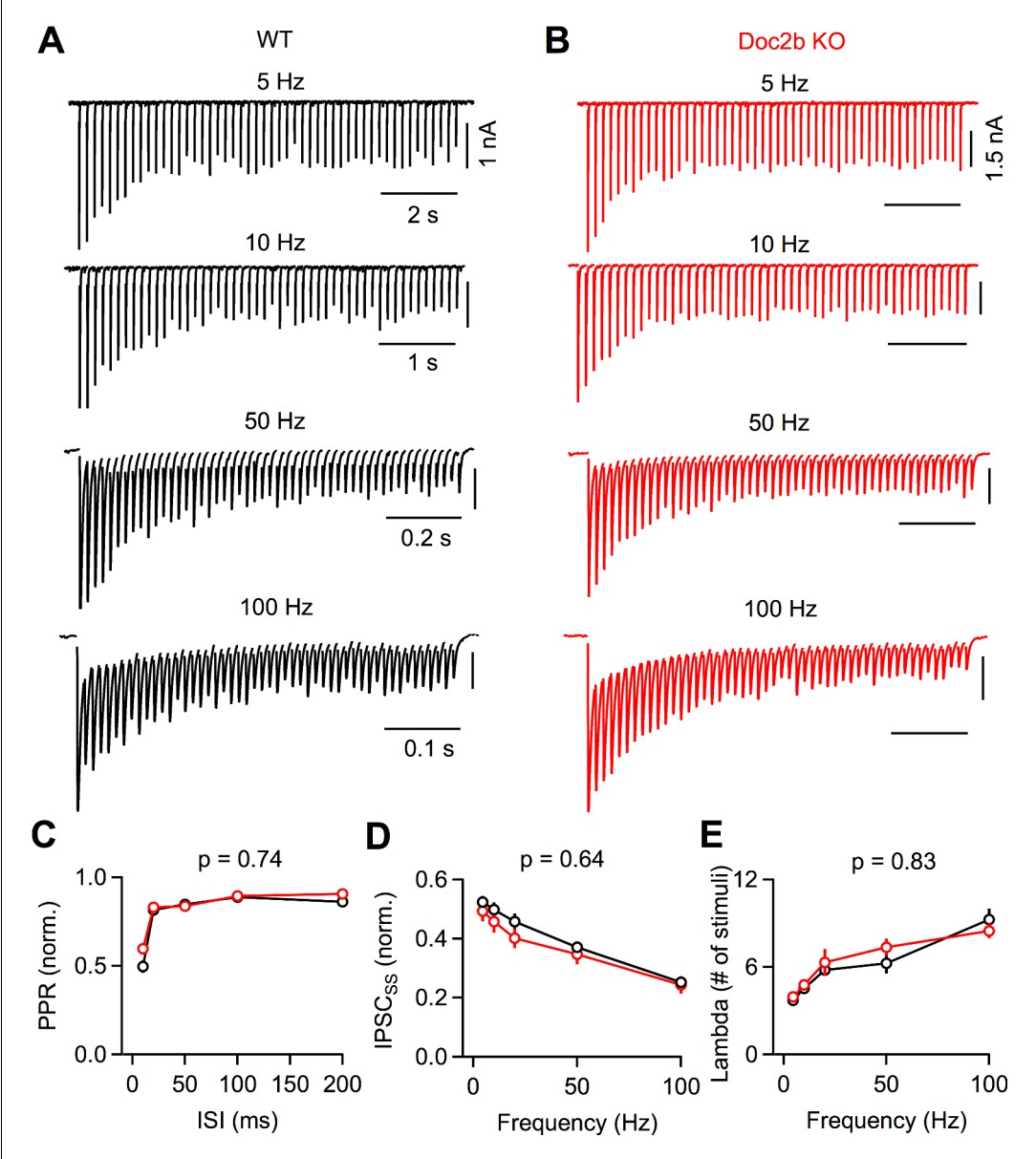

**Figure 3.** Short-term plasticity at the developing PC to DCN synapse is unaffected by Doc2bKO. (A) PC axons were stimulated at various frequencies and responses were recorded from large DCN neurons in P13-15 mice. Representative IPSCs for WT are shown with stimulus artifact blanked for clarity. (B) Same as in (A) but for Doc2b KO. Same scale as in (A) unless indicated otherwise. (C) Normalized paired pulse ratio (PPR) as a function of stimulation frequency. Data are mean ± S.E.M. n = 14 cells (6 animals) for WT, n = 11 cells (6 animals) for Doc2b KO. Statistical significance was assessed using two-tailed Student's *t*-tests after data were found to be normally distributed (Shapiro-Wilk test). (D) Same as in C but showing steady-state IPSC amplitude as a function of stimulation frequency. (E) Same as in C but showing steady-state decay time constant (λ) as a function of stimulation frequency.

The online version of this article includes the following source data for figure 3:

**Source data 1.** PC to DCN train data for young animals.

(p=0.74, two-tailed unpaired Student's *t*-test) (*Figure 3C*), suggesting that Doc2b does not contribute to the initial probability of release. We found no difference in the magnitude of the steady-state IPSCs between WT and Doc2b KO (p=0.64) (*Figure 3D*), and the kinetics of reaching steady-state were unaltered in Doc2b KO mice (p=0.83) (*Figure 3E*). These results demonstrate that Doc2b does not play a role in synaptic transmission during ongoing activity in juvenile PC to DCN synapses.

In adults, steady-state transmission at PC to DCN synapses is frequency-independent (*Turecek et al., 2017*; *Turecek et al., 2016*). This unusual property requires a precise balance between activity-dependent vesicle depletion and activity-dependent facilitation mediated by Syt7 (*Turecek et al., 2017*). However, synaptic responses are present during sustained high-frequency activation in Syt7 KO mice, although they are no longer frequency-invariant (*Turecek et al., 2017*). Therefore, there must be Syt7-independent mechanisms that help sustain release after repetitive PC activity. We hypothesized that Doc2b, which exhibits similar Ca-binding properties as Syt7, could help enhance release in response to PC activity. We tested this hypothesis by studying PC to DCN synapses from adults (P60-P80). Frequency-invariance is readily observed at this age in WT, as steady-state synaptic strength remains constant across PC firing frequency (*Figure 4A*; *Turecek et al., 2016*). In Doc2b KO mice, we observed no difference in synaptic responses compared to WT mice, and frequency-invariance remained intact (*Figure 4*). Specifically, steady-state synaptic strength and the kinetics of reaching steady-state were normal in Doc2b KO (p=0.32 and p=0.51, respectively) (*Figure 4C–D*). In addition, we examined whether the amount of Syt7-dependent facilitation was affected by Doc2b KO. If Syt7 and Doc2b are activated by a shared Ca source,

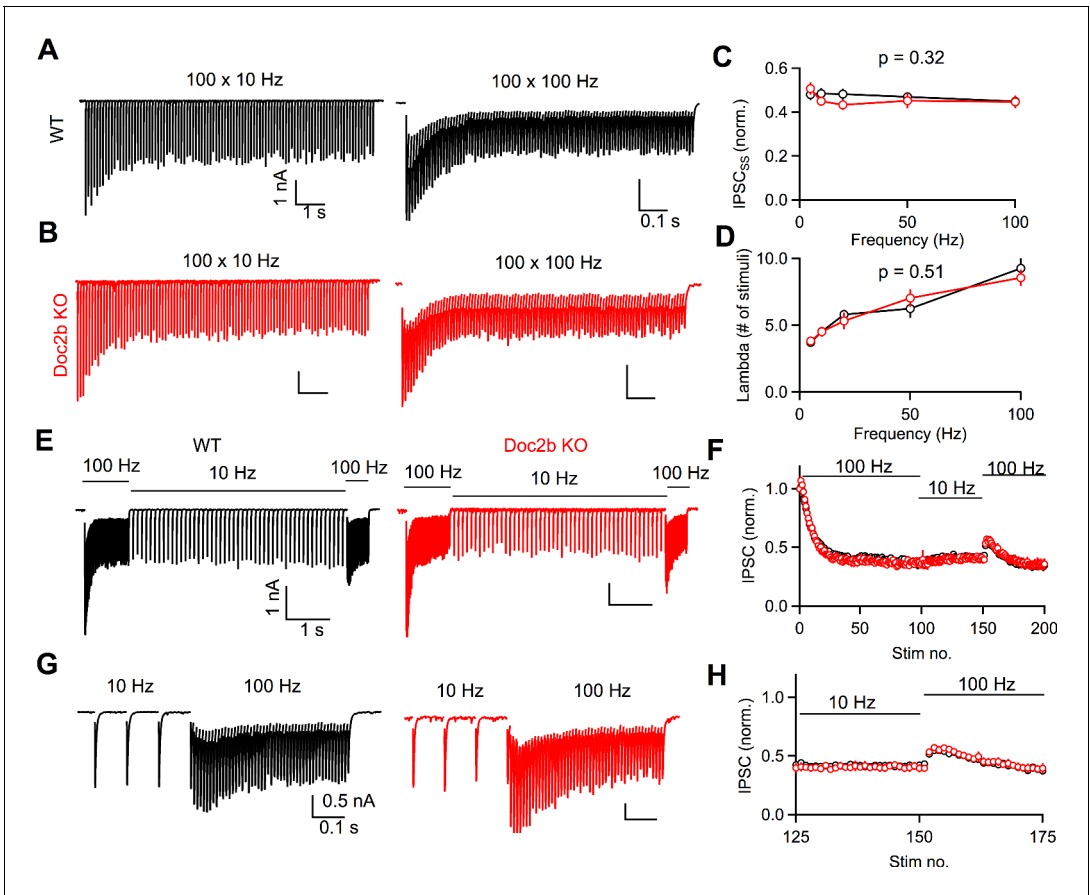

**Figure 4.** Doc2b is not necessary for frequency-invariant transmission at the mature PC to DCN synapse. (**A**) Example of DCN response to PC axon stimulation at 10 Hz or 100 Hz in WT. Stimulus artifact blanked for clarity. (**B**) Same as in (**A**) but for Doc2b KO. Same scale as in (**A**) unless indicated otherwise. (**C**) Normalized steady-state IPSC amplitude as a function of stimulation frequency. Data are mean ± S.E.M. n = 10 cells (four animals) for WT, n = 9 cells (four animals) for Doc2b KO. Statistical significance was assessed using two-tailed Student's *t*-tests after data were found to be normally distributed (Shapiro-Wilk test). (**D**) Same as in C but showing steady-state decay time constant ($\lambda$) as a function of stimulation frequency. (**E**) Example of DCN response to PC axon stimulation switched between 100 Hz and 10 Hz in WT and Doc2b KO, showing the frequency-independence of steady-state. Scale on right is same as scale on left. (**F**) Normalized IPSC amplitudes as a function of stimulus number for the protocol in (**D**). (**G**) Expanded view of examples in (**D**), showing a hidden facilitation unmasked when stepping from 10 Hz to 100 Hz. Scale on right is same as scale on left. (**H**) Same as in (**E**) but expanded view of frequency step from 10 Hz to 100 Hz.

The online version of this article includes the following source data for figure 4:

**Source data 1.** PC to DCN train data for adult animals.

then Syt7-dependent facilitation may become more prominent in Doc2b KO as Doc2b competing for Ca is absent. However, we did not observe any difference in the magnitude or time-course of facilitation in Doc2b KO at the PC to DCN synapse (*Figure 4G–H*). Altogether, our data do not support a role for Doc2b in PC to DCN synaptic transmission or short-term plasticity in young or mature animals under physiological conditions.

## Discussion

In this study, we aimed to answer whether Doc2b contributes to neurotransmitter release at a mature synapse. We used a hypothesis-driven approach that prioritized physiological relevance to clarify the function of Doc2b. We found no evidence to support previously proposed roles for Doc2b that were mainly based on studies in cultured cells. The frequency of spontaneous release was unaltered in Doc2b KO mice, indicating that Doc2b is not a calcium sensor for spontaneous release at this synapse (*Courtney et al., 2018*; *Groffen et al., 2010*; *Pang et al., 2011*; *Ramirez et al., 2017*). In addition, the PC to DCN synapse is extremely fast (average Tau decay = 2.1 ms; *Figure 2*), asynchronous release is not apparent in WT animals, and the time-course of synaptic responses is unaltered in Doc2b KO mice. We therefore conclude that Doc2b does not serve as a calcium sensor for asynchronous release (*Yao et al., 2011*) at PC to DCN synapses. Furthermore, the amplitudes of evoked synaptic responses were normal in Doc2b KO mice, suggesting that Doc2b does not influence the initial release probability or vesicle priming at the PC to DCN synapse (*Friedrich et al., 2008*; *Houy et al., 2017*). Finally, the observation that synaptic responses during prolonged high-frequency stimulation are maintained and unaltered in Doc2b KO mice, ruled out our hypothesis that Doc2b helps regulate transmission during sustained activation of the PC to DCN synapse (*Pinheiro et al., 2013*). Thus, we find that Doc2b does not contribute to any aspect of synaptic transmission at the mature PC to DCN synapse under physiological conditions.

It is surprising that the loss of this high-affinity calcium-binding protein which interacts with key secretory elements, including SNARE proteins and cell membranes (*Groffen et al., 2010*; *Pang et al., 2011*; *Yao et al., 2011*), fails to affect neurotransmitter release at the mature PC to DCN synapse. It is important to note that the studies implicating Doc2 in so many aspects of synaptic transmission were primarily performed at synapses between cultured cells. The only data previously reported for intact synapses in brain slices were conducted in very young (P7-8) mice at PC to PC synapses, where Doc2b KO reduces mIPSC frequency (*Groffen et al., 2010*; *Figure 1—figure supplement 3B*). It is not feasible to follow this effect over time because PCs do not provide the dominant source of inhibition onto other PCs after this brief period. This raises the possibility that the involvement of Doc2 in transmission could be a developmental effect on synaptic maturation. If true, this effect appears to be negligible by P11 for PC synapses, as mIPSC frequency was normal at this age and into adulthood at the PC to DCN synapse in Doc2b KO animals (*Figure 1*). It is also possible that redundant mechanisms and compensation by other calcium-binding proteins masked the effect of Doc2b KO. Although we have shown that this is not a consequence of compensation by Doc2a, which is absent from PCs in WT and Doc2b KO mice, there are many other calcium-binding proteins present in presynaptic terminals. One thing is very clear, however, loss of Doc2b does not influence neurotransmitter release at the mature PC to DCN synapse. We conclude that the roles of Doc2b in synaptic transmission described in previous in vitro studies do not necessarily apply to mature synapses under physiological conditions.

## Materials and methods

**Key resources table**

| Reagent type (species) or resource | Designation | Source or reference | Identifiers | Additional information |
|---|---|---|---|---|
| Gene (*M. musculus*) | *Doc2b* (gene) Doc2b (protein) | UniProtKB | P-70169 | |

*Continued on next page*

*Continued*

| Reagent type (species) or resource | Designation | Source or reference | Identifiers | Additional information |
|---|---|---|---|---|
| Strain, strain background (*M. musculus*) | Doc2b KO mice | DOI: http://doi.org/10.1126/science.1183765; PMID:20150444 | | |
| Antibody | Rabbit polyclonal anti-Doc2b | Synaptic Systems | Cat #174 103; RRID:AB_2619874 | IHC (1:200) |
| Antibody | Mouse monoclonal anti-Parvalbumin | Sigma-Aldrich | Product# P3088-.2ML; RRID:AB_477329 | IHC (1:500) |
| Antibody | Goat anti-rabbit IgG H and L Alexa Fluor647 | Abcam | Ab150083; RRID:AB_2714032 | IHC (1:1000) |
| Antibody | Goat anti-mouse IgG H and L Alexa Fluor568 | Abcam | Ab175473 | IHC (1:1000) |
| Sequence-based reagent | Fluorophore-conjugated Probe-Mm-Doc2b | ACD Bio | Cat#484798 | |
| Sequence-based reagent | 2.5 VS Probe - Mm-Doc2A probe | ACD Bio | Cat#531549-C2 | |
| Software, algorithm | Matlab | Mathworks (https://www.mathworks.com/downloads/) | RRID:SCR_001622 | Version R2017a |
| Software, algorithm | IgorPro | Wavemetrics (https://www.wavemetrics.com/order/order_igordownloads6.htm) | RRID:SCR_000325 | Version 6.37 |
| Software, algorithm | ImageJ software | ImageJ (http://imagej.nih.gov/ij/) | RRID:SCR_003070 | |
| Software, algorithm | OlyVIA software | Olympus (https://www.olympus-lifescience.com/en/support/downloads/) | RRID:SCR_016167 | Version 2.9.1 |
| Sequence-based reagent | Doc2b primers | This paper | PCR primers | 5'CATTGCCACTTCATAAGCGTAAGTTTCC 3' 5'CGAGGATGGAACCCTGTTTACTCTGG 33' 5'CCTTCTATCGCCTTCTTGACG 3' |
| Chemical compound, drug | NBQX disodium salt | Abcam | Ab120046 | |
| Chemical compound, drug | (R)-CPP | Abcam | Ab120159 | |
| Chemical compound, drug | Strychnine hydrochloride | Abcam | Ab120416 | |
| Chemical compound, drug | SR95531 (gabazine) | Abcam | Ab120042 | |
| Chemical compound, drug | Tetrodotoxin citrate | Abcam | Ab120055 | |
| Other | DAPI stain | Invitrogen (ThermoFIsher Scientific) | Cat#00-3958-02 | (1 µg/mL) |

## Ethics

All animal procedures were carried out in accordance with the NIH and Animal Care and Use Committee (IACUC) guidelines and protocols approved by the Harvard Medical Area Standing Committee on Animals (animal protocol #1493).

## Animals

Doc2b heterozygotes were kindly given by the Chapman laboratory and bred to produce Doc2b KOs and WT animals for experiments. Mice were originally produced in a C57/BL6 background (*Groffen et al., 2010*). Animal genotypes were assessed by PCR.

## Slice preparation

WT or Doc2b KO mice of both sexes were used for physiology experiments. Animal age varied across experiments (P12 and P40-P50 for PC to DCN mIPSCs, P7-8 for PC to PC mIPSCs, P18-P20 for PC to DCN minimal stimulation and burst stimulation, and P13-15 and P60-P80 for PC to DCN train experiments). Animals older than P20 were anesthetized with ketamine/xylazine/ acepromazine and transcardially perfused with warm choline-ACSF solution containing in mM: 110 Choline Cl, 2.5 KCl, 1.25 $NaH_2PO_4$, 25 $NaHCO_3$, 25 glucose, 0.5 $CaCl_2$, 7 $MgCl_2$, 3.1 Na-Pyruvate, 11.6 Na-Ascorbate, 0.005 NBQX, and 0.0025 (R)-CPP, oxygenated with 95% O2/5% CO2. To prepare sagittal slices of the cerebellum, the hindbrain was first removed, a cut was made down the midline of the cerebellum, and the two halves of the cerebellum were glued down to the slicing chamber. 150–200 µm thick sagittal slices were cut with a Leica 1200S vibratome in warm choline-ACSF. Slices were transferred to a standard ACSF solution containing, in mM: 127 NaCl, 2.5 KCl, 1.25 $NaH_2PO_4$, 25 $NaHCO_3$, 25 glucose, 1.5 $CaCl_2$, and 1 $MgCl_2$ maintained at 34–35°C for 10–12 min and then moved to room temperature for 20–30 min before beginning recordings. Procedures involving animals were approved by the Harvard Medical Area Standing Committee on Animals.

## Electrophysiology

Whole-cell voltage clamp recordings were performed on spontaneously active, large diameter (20–25 µm) neurons in the lateral and interposed deep cerebellar nuclei. These large DCN neurons have been characterized as glutamatergic projection neurons (*Uusisaari et al., 2007*). Selection criteria used to identify these cells are similar to those used in a previous study (*Turecek et al., 2016*), which confirmed glutamatergic identity of DCN cells in mice expressing TdTomato-labeled vesicular glutamate transporter (VGLUT2) (Slc17a6-IRES-Cre;Ai14). Cells were also selected in the more dorsal areas of the DCN along fiber tracts, as finding reliable PC inputs was easiest in these areas. Borosilicate glass electrodes were filled with a high chloride ($E_{Cl}$ = 0 mV) internal containing in mM: 110 CsCl, 10 HEPES, 10 TEA-Cl, 1 $MgCl_2$, 4 $CaCl_2$, 5 EGTA, 20 Cs-BAPTA, 2 QX314, and 0.2 D600, adjusted to pH 7.3 with CsOH. BAPTA was used in high concentration to prevent long-term plasticity (*Ouardouz and Sastry, 2000*; *Pugh and Raman, 2006*; *Zhang and Linden, 2006*). Low resistance (1–2 MΩ) electrodes were used to minimize series resistance (1–8 MΩ), which was compensated up to 80%. Compensation was only applied for the estimated capacitance of the cell body (5 pF). DCN cells were held at −30 to −40 mV and PCs were held at −60 to −70 mV. Liquid junction potentials were left unsubtracted. All experiments were done at 34–35°C. For DCN and PC recordings, 5 µM NBQX to block AMPARs, 2.5 µM (R)-CPP to block NMDARs, and 1 µM strychnine to block glycine receptors were included in the bath. For parallel fiber to PC recordings, 5 µM SR95531 was added in the bath to isolate mEPSCs. All experiments measuring mIPSCs or mEPSCs included 1 µM TTX in the bath. Flow rate was measured as 5 mL/min. All recordings and analysis were performed blind.

## Analysis

Recordings were obtained using Multiclamp 700B (Molecular Devices), sampled at 20 kHz and filtered at 4 kHz, and collected in Igor Pro (Wavemetrics). Data were analyzed using custom-written scripts in Matlab (Mathworks), which are available at https://github.com/waderegehr-lab/Doc2b-eLife (copy archived at https://github.com/elifesciences-publications/Doc2b-eLife; *Regehr, 2020*). All data are shown as means ± SEM unless otherwise indicated. For data obtained from each electrophysiology experiment, a Shapiro-Wilk test with significance level of 0.05 was used to test whether data were normally distributed. Most data were found to be normally

distributed and subsequently analyzed by a two-tailed unpaired Student's *t*-test. Some data were asymmetrically distributed with a right skew (*Figure 2*; *Figure 1—figure supplement 3*) and were tested by non-parametric Wilcoxon signed-rank. The threshold for statistical significance was set at $p < 0.05$.

The frequencies of miniature postsynaptic currents and spontaneous events were measured using the second derivative of the original trace to detect each event. Amplitudes were detected using an integration threshold. As DCN neurons have high spontaneous activity both with and without TTX, it was critical to make sure our event detection algorithm was sensitive and reliable. Therefore for each cell, we checked the output of the detection algorithm by plotting individual traces along with each event detected by the algorithm and visually ensured that events were detected for each observed event.

During trains, IPSC amplitudes were measured from averaged traces as the peak evoked current relative to a baseline measured 2 ms before onset of the stimulus. In young animals, IPSCs usually do not fully decay during high-frequency trains before subsequent stimuli. To accurately measure IPSC amplitude, baselines were then measured by extrapolating a single exponential fit from the previous IPSC. The steady-state IPSC was measured as the average size of the IPSCs between the 50th to 80th stimuli.

## Fluorescence in situ hybridization (FISH)

Eight- to 9-week-old WT and Doc2b KO animals were anesthetized with isoflurane before the brain was quickly removed, frozen in dry ice, and embedded in optimal cutting temperature (OCT) compound (Tissue-Tek). 20-µm-thick sagittal slices of the whole brain were cut on a cryostat (Microm HM500-CM) and mounted on glass slides (Superfrost Plus, VWR, 48311–703). Fluorescent in situ hybridization was carried out according to the ACD-Bio RNAscope Multiplex Assay manual (document Number 320514). Samples were subsequently fixed in 4% paraformaldehyde in phosphate-buffered saline (PBS) for 15 min at 4°C and then dehydrated with 50% (x1), 70%, and 100% (x2) ethanol washes for 5 min each. Slides were air-dried and a barrier around the tissue was drawn using an Immedge hydrophobic barrier pen (Vector Laboratories). Brain slices were then incubated in RNA-scope protease III reagent (ACD-Bio 322337) at room temperature for 30 min and rinsed twice in PBS for 5 min. Fluorophore-conjugated Probe-Mm-Doc2b (Cat #484798) and 2.5 VS Probe -Mm-Doc2A probe (Cat# 531549-C2) were incubated with the slide-mounted tissue sections in a HybEz II oven (ACD-Bio) for two hours at 40°C, then washed in RNAscope wash buffer reagent (ACD-Bio 310091) twice. To amplify fluorescence signals, the tissue was incubated in AMP 1-Fl for 30 min at 40°C (HybEZ oven), washed twice with wash buffer for 3 min at room temperature. Subsequently, the tissue was incubated in AMP 2-FL for 15 min at 40°C, washed twice, and incubated in AMP 4-LA-A for 15 min at 40°C, then washed twice again. Sections were then stained with DAPI and mounted using ProLong antifade reagent (Thermo Fisher Scientific P36930). Positive control probes to ensure in situ quality included housekeeping genes (C1-Mm-Polr2a, C2-Mm PPIB, and C3-Mm-UBC). Fluorescence from negative control probes targeting bacterial RNA (C1, C2, C3-dapB) was not detected. Slides were imaged at 20x (air) by a whole slide scanning microscope (Olympus VS120). All image acquisition and processing were done blind.

## Immunohistochemistry and imaging

Mice were anesthetized with isoflurane and perfused first with cold phosphate buffered saline (PBS, pH = 7.4, Sigma Cat# P-3813), then by 4% paraformaldehyde in PBS. The brain was removed and post-fixed overnight at 4°C in the same solution. For slicing, the brain was embedded in 4% agar (Sea Plaque, Lonza, Cat# 50101) and sliced in PBS using a vibratome (VT1000S, Leica) at a thickness of 50 µm. Slices were then incubated in blocking solution containing 4% normal goat serum (NGS) in PBS for 1–2 hr. After blocking, the slices were incubated in the same solution with the addition of primary antibody overnight at 4°C (rabbit polyclonal anti-Doc2b (1:200; Synaptic Systems, Cat# 174 103) and mouse monoclonal anti-Parvalbumin (1:500, Sigma-Aldrich, Product # P3088-.2ML)). Slices were then washed three times for 10 min. Next, slices were incubated in 4% NGS and 0.1% triton X-100 in PBS with the addition of secondary antibodies for 2 hr at room temperature (goat anti-rabbit IgG H and L Alexa Fluor647 (1:1000, ab150083) and goat anti-mouse IgG H and L Alexa Fluor568 (1:1000, ab175473)). Slices were then washed three times for 5 min in PBS, mounted on glass slides

and covered with mounting medium (Invitrogen Fluoromount-G Mounting Medium, Cat #: 00-4958-02) and a glass coverslip. Mounting medium was allowed to cure for at least 24 hr before imaging.

Whole-brain images were taken on an Olympus VS120 slide scanner (*Figure 1—figure supplement 2*), and confocal stacks were acquired on an Olympus FV1000 confocal microscope (*Figure 1C–D*). Images were acquired and processed using standard routines in Fiji (ImageJ) using identical settings across genotypes. All image acquisition and processing were done blind.

## Acknowledgements

We thank Pascal Kaeser for valuable help and insightful discussion. We thank Edwin Chapman (University of Wisconsin – Madison) for providing Doc2b KO mice. We thank Mahmoud el-Rifai and the Harvard Medical School Neuroimaging Core for help with RNAscope and Stephanie Rudolph and Christopher Chen for help with immunostaining. This work was supported by National Institutes of Health Grant R35NS097284 to WGR and a National Science Foundation Graduate Research Fellowship under grant DGE1745303 to MMK.

## Additional information

### Funding

| Funder | Grant reference number | Author |
|---|---|---|
| NIH Office of the Director | R35NS097284 | Wade G Regehr |
| National Science Foundation | DGE1745303 | Mehak Khan |

The funders had no role in study design, data collection and interpretation, or the decision to submit the work for publication.

### Author contributions

Mehak M Khan, Conceptualization, Data curation, Investigation, Writing - original draft, Writing - review and editing; Wade G Regehr, Conceptualization, Resources, Funding acquisition, Writing - original draft, Project administration, Writing - review and editing

### Author ORCIDs

Mehak M Khan ![ORCID] https://orcid.org/0000-0001-5710-7421
Wade G Regehr ![ORCID] https://orcid.org/0000-0002-3485-8094

### Ethics

Animal experimentation: All experiments were conducted in accordance with federal guidelines and protocols (#1493) approved by the Harvard Medical Area Standing Committee on Animals.

### Decision letter and Author response

Decision letter https://doi.org/10.7554/eLife.55165.sa1
Author response https://doi.org/10.7554/eLife.55165.sa2

## Additional files

### Supplementary files

• Transparent reporting form

### Data availability

All data used in this study are originally generated. Source data are included in the source data files indicated for each Figure or Figure supplement. Values of individual measurements within an experiment are included in captions for relevant graphs.

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
