## [Decision Letter]

**Acceptance summary:**

All reviewers, including me, agree that your study represents an important contribution to our understanding of presynapse function and that it nicely highlights the requirement to study molecular and cellular mechanisms of presynapse function in multiple preparations in order to reach meaningful conclusions with lasting relevance.

**Decision letter after peer review:**

Thank you for submitting your article "Loss of Doc2b does not influence transmission at a mature synapse under physiological conditions" for consideration by *eLife*. Your article has been reviewed by three peer reviewers, and the evaluation has been overseen by a Reviewing Editor and Richard Aldrich as the Senior Editor. The following individual involved in review of your submission has agreed to reveal their identity: Rafael Fernández Chacon (Reviewer #1).

The reviewers have discussed the reviews with one another and the Reviewing Editor has drafted this decision to help you prepare a revised submission.

Summary

All three reviewers note that DOC2s are fascinating presynaptic calcium-binding proteins, and acknowledge that, despite substantial efforts, the exact role of DOC2s in synapses within intact circuits is essentially unknown. Correspondingly, they all appreciate your effort to address this important issue by testing which of the previously proposed roles of DOC2s (i.e. regulating spontaneous vesicle fusion, asynchronous vesicle fusion, augmentation, priming) can be validated in a defined synapse in brain slices.

The study, which is based on the PC-to-DCN synapse as a model, is beautifully conducted and comprehensive. The striking conclusion is that none of the relevant characteristics of mature PC-to-DCN synapses are affected by genetic deletion of DOC2s – specifically of DOC2b, which is very likely (based on available evidence) the only DOC2 expressed in these synapses. This is an important finding of substantial importance to the field of synapse biology.

Major comments

The following two issues should be addressed before the paper can be accepted for publication in *eLife*:

1) You assume that DOC2b mRNA levels are a faithful proxy of DOC2b protein expression, but this cannot be taken for granted (e.g. due to scenarios such as high DOC2b degradation rates or poor synaptic trafficking). It is important that you show, e.g. by using immunostaining, that DOC2b is indeed present at PC-to-DCN synapses, how its levels at PC-to-DCN synapses compare to those at other synapse types, and that the corresponding staining is gone in the knock-out. If DOC2b levels were intrinsically very low at PC-to-DCN synapses, the present study might not be all that informative.

2) It is striking that PC-to-DCN synapses are not affected by DOC2b knock-out, and the corresponding “negative” dataset is important. However, in some parts of the present manuscript, the corresponding conclusions are phrased too categorically. For instance, at the end of the Abstract, you state "that conclusions based on cultured cells do not generalize to mature synapses under physiological conditions". Similar sentences appear at the end of the Results section and again at the end of the Discussion section. This does not match with the fact – which you concede in other parts of the manuscript – that the present data only concern one synapse type. Given that many studies, in cultured cells and in more intact preparations alike, showed that defined presynaptic proteins affect different synapse types differently, a more scholarly and careful phrasing and a corresponding extension of the relevant parts of the Discussion are necessary. This could be done at the end of the Discussion, where some caveats, such as alternative sensor proteins, are already mentioned. Also, the synapse type studied should be mentioned in the title and Abstract. In essence, key aspects of DOC2 function may be relevant during development and/or play a major role in synapse types that were not studied here.

---

## [Author Response]

Major commentsThe following two issues should be addressed before the paper can be accepted for publication in eLife:1) You assume that DOC2b mRNA levels are a faithful proxy of DOC2b protein expression, but this cannot be taken for granted (e.g. due to scenarios such as high DOC2b degradation rates or poor synaptic trafficking). It is important that you show, e.g. by using immunostaining, that DOC2b is indeed present at PC-to-DCN synapses, how its levels at PC-to-DCN synapses compare to those at other synapse types, and that the corresponding staining is gone in the knock-out. If DOC2b levels were intrinsically very low at PC-to-DCN synapses, the present study might not be all that informative.

This was an excellent suggestion. We have done immunostaining to show that Doc2b is present at PC to DCN synapses and that Doc2b is not detected in the Doc2b KO (Figure 1 C-D). We include additional immunostaining, including a characterization of the hippocampus in Figure 1—figure supplement 2.

2) It is striking that PC-to-DCN synapses are not affected by DOC2b knock-out, and the corresponding “negative” dataset is important. However, in some parts of the present manuscript, the corresponding conclusions are phrased too categorically. For instance, at the end of the Abstract, you state "that conclusions based on cultured cells do not generalize to mature synapses under physiological conditions". Similar sentences appear at the end of the Results section and again at the end of the Discussion section. This does not match with the fact – which you concede in other parts of the manuscript – that the present data only concern one synapse type. Given that many studies, in cultured cells and in more intact preparations alike, showed that defined presynaptic proteins affect different synapse types differently, a more scholarly and careful phrasing and a corresponding extension of the relevant parts of the Discussion are necessary. This could be done at the end of the Discussion, where some caveats, such as alternative sensor proteins, are already mentioned. Also, the synapse type studied should be mentioned in the title and Abstract. In essence, key aspects of DOC2 function may be relevant during development and/or play a major role in synapse types that were not studied here.

We adjusted the title, Abstract, the end of the Introduction and the end of the Discussion.